# Coach Competences to Induce Positive Affective Reactions in Sport and Exercise—A Qualitative Study

**DOI:** 10.3390/sports7010016

**Published:** 2019-01-08

**Authors:** Ulrich Georg Strauch, Hagen Wäsche, Darko Jekauc

**Affiliations:** 1Institute of Sport Science, Humboldt-University of Berlin, 10115 Berlin, Germany; 2Institute of Sports and Sports Science, Karlsruher Institute of Technology, 76131 Karlsruhe, Germany; hagen.waesche@kit.edu (H.W.); darko.jekauc@googlemail.com (D.J.)

**Keywords:** coach’s behaviour, affective reactions, physical activity, behaviour-related competences

## Abstract

Positive affective reactions are a crucial aspect in physical activity maintenance. Affective reactions to sport and exercise were found to be important factors of physical activity. Coaches could be an important medium to induce positive affective reactions of participants in sport and exercise. Understanding how coaches trigger positive affective reactions (AR) during physical activity is a crucial aspect for increasing maintenance in sport and exercise. The aim of this study is to identify the competences of the coaches which are associated with perceived positive AR of participants during sport and exercise. To identify these factors, semistructured in-depth interviews were conducted with 18 participants, who take part in sport and exercise (nine female and nine male) of heterogeneous age (mean age 42.6; SD = 19.25; under 30 years, 30 to 60 years, 60 years and above) and who have different athletic backgrounds (individual sports, team sports, and gym classes). Four key coach competence factors were identified and used to design an integrated model. Three general competences: context sensitivity, social–emotional competences, professional competences, and the specific competences in the behaviour-related competences.

## 1. Introduction

Participation in physical activity is associated with a variety of health benefits and a reduction of chronic diseases [1]. Regularly skipping or substantially reducing physical activity can lead to a loss of any initial health improvement [2,3]. To maintain the benefits of physical activity, continuous and regular participation in physical activities is crucial [4,5,6]. An enormous number of participants in sport and exercise programs drop out during the first six months [7,8]. So far, only little research has examined the aspects of continued maintenance of physical activity [9]. Affective states such as emotions and affects have been recently discussed as determinants of physical activity maintenance [10].

According to Baumeister [11], automatic affects can be regarded as “automatic reactions (such as liking or disliking something) that are simple and rapid”. These reactions can be conscious or unconscious and are typically “no more than a quick twinge of feeling that something is good or bad, of liking or disliking for something”. Russel [12] postulates a similar concept with the term “core affect” which he defines as a neurophysiological state, accessible to consciousness as a simple nonreflective feeling: “…feeling good or bad, feeling lethargic or energised”. On the contrary, emotions represent certain fully developed, consciously felt and differentiated emotional states that are associated with cognition [11]. Clore and Ortony [13] call for a shift from discussing “emotion” to discussing “affective processes”. In this study, the term affective reactions (AR) is used which comprises both affects as well as emotions.

### 1.1. Influence of Affective Reactions on the Maintenance of Physical Activity

To this day, there is considerable empirical evidence suggesting that AR are important predictors of physical activity. The review of Rhodes, Fiala and Conner [14] suggests that many studies have consistent and significant correlations between AR and physical activity. For example, Kiviniemi [15] showed that affective attitude is a more important determinant of physical activity than cognitive attitude. Furthermore, a more recent review by Rhodes and Kates [16] revealed that AR are consistently related to future physical activity. Obviously, participants who enjoyed physical activity more had a higher probability to stay physically active compared to those who did not enjoy physical activity. In an experimental study over 20 weeks, Jekauc [17] showed that an intervention to increase the enjoyment of physical activity also increases the tendency to maintain being physically active. Affective neuroscience evidence also indicates that affective processes are crucial in the decision for, or against certain behaviours (e.g., empathy or context sensitivity of coaches in physical activity) [18].

### 1.2. Coach Competences and Affective Reactions

In the context of sports and physical activity, the coach seems to be an important agent influencing the AR of participants [19,20]. In this study, the term coach is used as an overarching term for all sport coaches and fitness leaders for recreational athletes and exercisers in nonprofessional sports. According to Borggrefe, Thiel and Cachay [21] coaches nowadays not only need to utilize subject specific know how when designing a sport and exercise classes structure, but also their social abilities. Although coaches with ample experience intuitively use specific strategies to positively influence the AR of sport and exercise participants, there seems to be little systematic knowledge on how they actually achieve this result. One specific strategy, to positively influence the AR of sport and exercise participants, could be to improve the competence to motivating the participants individually when being physically active. This idea is supported by a study by Strauch et al. [22] in which it was found that specific and general coach competences can influence AR of participants in sport and exercise and result in increased sport participation meaning higher commitment of sport participants. Such a competence could be the coach’s ability to establish a good relationship with the participants. The results of a study by Olympiou, Jowett, & Duda [23] also point towards a correlation between the coach–athlete relationship and the motivation of participating athletes. It was shown that the athlete’s perceptions of their specific relationship with the coach had a significant correlation with the perceived motivation by the coach. Moreover, the influences of coach behaviour on athletes were also addressed by Chelladurai [24], based on the Multidimensional Model of Leadership. He identified several different components (e.g., interpersonal skills and empathy) which affect the coach and his/her behaviour. In this model, certain key antecedents (situational, leader and member characteristics) were addressed. It also included expected and desirable behaviour of a leader which was compared to the leader’s actual behaviour and its consequences (performance and satisfaction). Different components that influence the behaviour of coaches are also described in the Coaching Model by Coté, Salmela, Trudel and Baria [25]. Competition, training and organisational components which are also defined as the coaching process are the foundation of this model. Four variables affect the coaching process: the coach’s personal characteristics, the athletes’ personal characteristics, level of development and some contextual factors [25]. 

In this context, another factor of coach competences is leadership style. Two types of leadership styles and their influences on exercise-related outcomes have been discussed: bland leadership style and enriched leadership style [26]. Both styles address at a minimum what is required of a fitness professional; yet the enriched style of leadership exceeds expectations, in comparison to the bland instructor [27]. Martin and Fox [26] outlined the type of bland instructor as someone who avoids social interaction with the participants and does not promote interaction between the participants. These kinds of coaches do not want to receive more information than necessary; they are not interested in the names of the participants and do not want to give feedback nor praise. Generally speaking, this type of instructor tends to result in a negative experience for the participants [26]. An enriched coach takes care of his/her participants’ sensitivities. An enriched leadership style is represented by an instructor who is socially interactive, pleasant and energetic [26]. The coach provides positive encouragement and gives positive feedback on performances. Enriched instructors address participants by name, they are interested in general conversation (before and after class), give specific reinforcement and ignore mistakes whilst rewarding effort before and after sport and exercise [26]. Based on these findings, it is noteworthy that none of the studies included in the reviews of Rhodes, Fiala and Conner [14] and Rhodes and Kates [16] examined the role of the coaches and how their influence might work in this context. Up until now, this aspect has barely been examined in any study regarding AR.

Although several models have been postulated, it is little known which competences and behavioural strategies of the coaches influence AR and emotions of sport and exercise participants. Which competences do successful coaches need to have in order to induce positive AR of participants in sport and exercise programs? To answer this question, a qualitative interview study was conducted. The aim of this explorative study was to identify competences a coach needs to have to induce positive AR of participants during sport and exercise. To clarify the interaction of various aspects, the aim was to develop an integrated model that displays relevant competences and their interactions regarding AR of sport and exercise participants. 

## 2. Methods

To explore emotional processes, like affective reactions in sport, using a qualitative design is a very useful approach [28]. An open, qualitative approach enables a deeper understanding of individual perceptions and mechanisms of emotional processing. Adding to this, to date, little is known about which coach competences are associated with positive and which with negative AR of participants. There is no basis to deduce hypotheses for quantitative studies. Therefore, a qualitative research design seems suitable for exploring the relationship, for generating new hypotheses and promoting theory development. The quality of information gained from participants of sport and exercise through interviews, in which they share their feelings and emotions, is of considerable value. 

### 2.1. Participants

The interviews were conducted with 18 participants, who have all taken part in different sport and exercise programs. All participants exercised for recreational purposes and not for professional competitive sports. The study took place in Berlin, the capital and largest city in Germany with more than 3.5 million inhabitants. All participants were of German origin. To ensure heterogeneity of the sample, three different factors were taken into consideration: age (18–29 years vs. 30–59 years vs. over 60 years), sex (9 male vs. 9 female) and type of sport (6 individual vs. 6 team sport vs. 6 gym classes). Finally, for each of the 18 (3 × 2 × 3) possible factor combinations one participant was recruited. The average age of the participants was 42.6 years (SD = 19.25). The study was accepted by the ethics committee of the Goethe University of Frankfurt/Main (code 2018–51). In the interviews, there was no physical or psychological health risk for any of the participants (not vulnerable population). All participants had the possibility to leave the interview at any time and gave written consent before participating in the study.

### 2.2. Data Collection

The gym class participants were recruited from various gyms. Participants of team sports were recruited from soccer, handball and basketball teams and participants recruited from individual sports had backgrounds such as sailing, martial arts or weight training. The data collection was based on semistructured, in depth interviews, based on an interview guide. Participants were interviewed in separated, quiet rooms or next to places where they usually exercise or train. These interviews were conducted by three different people who had all received comprehensive interviewing training. During this training, important aspects of semistructured interviewing were taught, which leaves room for the participants’ own thoughts and comments, it could otherwise possibly leave out any important aspects for us. As the interviews were taking place a protocol was kept by another person in order to record specific occurrences (e.g., nonverbal communication or atmosphere). The interviews were recorded and later transcribed verbatim by the respective interviewer. In the interview, the participants were only asked questions formulated in an open-ended format. Questions concerned their experiences, habits and affects with regard to key competences of the coach’s behaviour, which were associated with perceived enjoyment whilst exercising. In the first part of the interview, nonspecific personal ‘ice-breaker’ questions were posed such as “what is your position in your current job and what are your tasks there?” In the second part, specific questions about sport and exercise classes, and the support given by the coach, such as “how did you become interested in this sport?”, and, “how would you like your coach to supervise you?”, were asked. The third part consisted of questions about the satisfaction with the coach’s support and specific characteristics like “what qualities should a coach offer in terms of behaviour and support?” or “what do you expect from a coach in order to make you feel good during sport?” In the final part of the interview, questions concerned affects perceived during sport and exercise, and related to the coach’s support. For example, “can you remember and explain a situation in which your coach supervised you excellently/badly?”, “what were your emotions and feelings like during the session supervised by the coach?” and “what caused these emotions and feelings in this situation?” The participants were told that there are no right or wrong answers and that the only important aspect is the personal experience of the participants during sport and exercise classes. The interviewer had no previous contact with the participants’ coaches and the data remained anonymous and was not passed on to the coaches.

### 2.3. Data Analysis

Data analysis took place based on the transcripts derived from the interviews. To analyse the qualitative data, the principles of the grounded theory [29] were applied. Since there is little research in this specific field of affective reactions in leisure sports and physical activity, the aim of the study was not to examine specific hypotheses through quantitative analysis, but to explore a new field of research to generate a theoretical framework that allows formulating hypotheses. In this regard, the principles of the grounded theory provide an appropriate analysis strategy [29]. The first step of constructing a theory is open coding. Here, data is broken down analytically. This involves the encoding of the entire text and grouping it into thematic parts with a clear subdivision of the main categories (e.g., professional expertise and social–emotional competence) and a few subcategories (e.g., relationship management and empathy) of coach competences. To identify and differentiate coach competences, specific characteristics of the competences were coded.

The next step included the axial coding. Here, the aim was a clear apportionment between the different aspects of the competences. Two researchers analysed relationships between the main categories and the specific subcategories. Also, in this process the categories were developed further and the researchers looked at further indicators [29]. In the final step, selective coding was applied. Here the focus lies on the core category and its dimensions as well as, the interactions between main and subcategories. This step includes a precise definition rather an explanation of the categories and the generation of hypotheses regarding the interactions of competences.

Other categories that needed further explanations were filled-in with descriptive details. The selective coding was done by three researchers. Based on the identified competences and their interaction, a model was constructed. 

## 3. Results

In the interviews, 46 different characteristics of coach competences were identified. The coaches’ characteristics that are associated with perceived positive AR of participants during leisure sports and physical activity were summarised and categorised, and interactions among the categories were analysed. The categories were defined and explained based on the participants’ statements in the interviews and their interpretation. This resulted in an integrated model (Figure 1) which summarises coach competences and their interaction. It comprises four distinct key categories. Three general competences—context sensitivity, social–emotional competences and professional competences—and the resulting specific competences: the behaviour-related competences. The competences interact, as can be seen in the model (Figure 1), and are indicated by the arrows. The behaviour-related competences are based on the general behaviour competences and describe specific abilities of coaches. The behaviour-related competences comprise related subcategories: motivation competence, adaptability competence and organisational competence. The model starts with the first general competence, the context sensitivity. The analysis of the context duly influences the social–emotional competences, especially empathy and the professional competences. Based on these general competences, which enable coaches to properly address different contexts, the specific competences, enable the coach to take actions.

All categories of the model refer to the specific and general competences of the coach towards the participants during leisure sports and physical activity. In the following, these categories and their interactions are described and discussed in detail based on the previously analysed interviews.

### 3.1. Description of the Coach Competences

#### 3.1.1. Context Sensitivity

The first main category in the model is context sensitivity. It influences all other coach competences. Context sensitivity is a general competence and represents the capability to recognize situational aspects that enable coaches to respond to specific conditions such as age, sex and personality as well as physical and social factors. A coach with high context sensitivity is able to recognize that with different social backgrounds, such as religion (e.g., Muslims and Christians), follow different social rules. Another example is the necessity to address younger and older people differently (e.g., through choice of words). Context sensitivity is strongly influenced by the coaches’ experiences. The statements in the interviews, with regards to the coach’s age, underlined that the more experience the coach has, the better his/her capability to recognize and analyse specific situations in sport and exercise. In almost all interviews, sport and exercise participants expressed the expectation that the coach adapts his/her behaviour and communication with regards to specific aspects of personal interaction. For example, a handball coach can place more demands on a young handball player than on older players, and the coach can motivate the young players with more communication for example using English terms, which older players would not necessarily understand (in German speaking countries). Pointing to differences in gender, a participant of handball mentioned: “It is also important that the coach can behave and talk differently to men in the training than to women. He must be able to switch, otherwise it does not work for long!” The recognition of the intricacies of a specific situation activates the selection of specific behaviour related to professional competences. This enables appropriate action according to the situation due to the coach’s social–emotional competences.

#### 3.1.2. Social–emotional Competences

The second main category is represented by social–emotional competences. These general competences represent the ability to develop a positive supportive and appropriate relationship with the participants. The social–emotional competences comprise emotional as well as social aspects which are mirrored in three subcategories: relationship management, self-management and empathy. These subcategories of social–emotional competences are described in the following. 

The first subcategory of the social–emotional competences is *relationship management*. This management skill represents the competence to manage the interaction with participants in order to develop an appropriate relationship between coaches and participants. It comprises communication, leadership and collaboration. Communication refers to clarity of instructions, unambiguous feedback, friendliness, being communicative, eloquence, politeness and coming across in a relaxed manner. For example, male participants from the 18–29 years old group considered it to be important, that the coach supports their sport and exercise class experience by commenting positively on their progress or suggesting methods to improve. In this context, the way in which the coaches communicated, for example, speaking in a friendly tone and no shouting, was also considered to be important. Leadership refers to assertiveness, fairness and dominance. The interviews showed that many participants with different athletic backgrounds have the desire to be led throughout the workout. The coach should operate as a leader in order to be able to apply his/her expertise. He/she should treat all athletes equally and should not, for example, differentiate between players in a sports team. Collaboration means to receive and reflect suggestions and criticism of participants as well as building a respectful and trustworthy relationship with them. As an example, male participants from the 30–59 years old group emphasised the importance of taking justified criticism seriously and adjusting the content of the sport and exercise classes or even their behaviour, accordingly. Furthermore, female participants of the same age group mentioned that the coach ought to deal with private conversational content sensitively and confidently. For example, a sailing participant stated: “I must be able to trust my coach with something personal and be sure that it will stay confidential and not be passed around.”

The second subcategory of the social–emotional competences is *empathy*. Empathy represents the competence to feel and understand another person’s situation from their perspective. Coaches who are empathetic are perceived as sympathetic and sensitive to the feelings of participants. For example, male participants in the over 60 years old group underlined how important it is that the coach is sensitive to the participants’ needs. A participant of spinal exercises said in the interviews:

… at an old age every old person has a few problems, for example my hip pain, which a coach must know about. My individual problem in this situation has to get recognised without me having to say something and he must react to my particular problem.

Working on a step-by-step basis without pressure was considered to be important, because it takes time to convince people or to remind them that they are exercising for their own sake and not for the sake of the coach. Empathy can be regarded as a prerequisite for relationship management.

The third subcategory of the social–emotional competences is *self-management*. This management skill is the competence to regulate one’s emotions in order to ensure an appropriate interaction with participants. It is expressed through seriousness, authenticity, dispassion, honesty, humility, staying attentive, having patience and self-control. For example, female participants referred to negative experiences where the coach used frantic and bad language or was dishonest and had a very subjective view of the sport and exercise class’s content. A soccer participant said in the interview: “Sometimes, I think of situations in football training, where the players are reviled by some coaches. I sometimes wonder whether they had received any education at all, or not? There is no reason for it!” Self-management can be regarded as a prerequisite for both empathy and relationship management. 

#### 3.1.3. Professional Competences

The third main category represents *professional competences*. These general competences reflect the professional and sports-specific oriented features of the coach’s behaviour. These competences comprise profound skill-specific knowledge as well as skills to correctly demonstrate the movements, exercises and game procedures, which are mirrored in two subcategories: motor competence and professional expertise.

The first subcategory is *motor competence*. This competence represents the motor skills that the coach requires to correctly demonstrate sport-specific movements, for example the different moves required in a gym class choreography, such as a V-Step in an aerobics class. Other examples include the exact execution of particular exercises on fitness equipment, such as a bench press, or precise technique in ball sports, such as a penalty kick in soccer. A participant of judo said in the interview: “The most important thing is that the coach is able to convince us with his performance that he can show us the way to new techniques and that the coach can show it correctly!” This is an important way to learn, because participants can use the coach’s demonstration and imitate it. Motor competence facilitates learning in sports. A coach with high motor competence earns the respect of the participants.

The second subcategory of the professional competences is *professional expertise*. It refers to skill-specific knowledge, which has been acquired in sport-specific education and own sport-specific experiences. Coaches with high professional expertise have the knowledge to correct, instruct and control the actions of the participants properly. For example, a female participant in a gym class group, aged over 60 years, considered it to be of major importance that the coach possesses a lot of expertise and know how to exercise correctly. According to one of the participants, this expertise and know how is an advantage in comparison to other coaches. A participant of a body combat course said in the interview: “It’s just important, as I’ve just said that he’s competent and knows what he’s doing! He knows how and when he corrects us, it means we do not do anything wrong and injure ourselves.”

#### 3.1.4. Behaviour-Related Competences

The behaviour-related competences describe specific abilities of coaches to create and carry out a tailored program, considering social as well as content-related issues in sport and exercise. These competences base on general competences such as context sensitivity, social–emotional competences and professional competences. The behaviour-related competences serve as immediate factors that influence the coach’s behaviour. Behaviour-related competences are directly related to AR of participants in sport and exercise programs. The behaviour-related competences consist of three specific competences: motivation competence, adaptability competence and organisational competence. 

The first subcategory of the behaviour-related competences is the motivation competence. The coach strives to motivate the participants by animating, praising and generating fun. He/she often entertains the participants in order to motivate them. By doing this, the coach creates a positive and motivating atmosphere. For example, female participants from the 18–29 years old group described situations in which the coach took part in sport and exercises. He/she counted the repetitions aloud, praised the performance of the participants and motivated them to do even more. A participant of the step course (exercise class) said in the interviews: “I think quite often: I cannot do any more, I would like to stop immediately, but then my coach comes and stands next to me and might do the last push-up with me and this motivates me to continue.” Another example of a coach motivation competence was described by a basketball class participant: “I always need a boost of motivation. I’m just lazy at the moment and tend to quit things very quickly. I just need the motivation of the coach or sometimes a ‘you do well’ or ‘here you can improve even more’. I just need the support.”

The behaviour-related competence—adaptability competence—represents an integrative competence to tailor the content of the sport and exercise classes and the coach’s ability to adapt his/her behaviour to the participants’ needs. It integrates influences of the professional competences, the social–emotional competences and especially the context sensitivity. The adaptability competence enables the coach to develop a specific regime of sport and exercise classes which corresponds with the needs of the participants and social context of the situation. This is shown through highly efficient, customised content, a varied program, personal judgement and specific vocabulary. The interviews show that coaches have to adapt to their audience. This means, that the sport and exercise classes’ content and the coach’s behaviour need to be very different according to the target group. For example, the needs of elderly women should be differently catered for than those of younger men. Every audience needs specific and individual styles of sport and exercise classes with individual choice of words and specific contents in order to induce positive AR. A participant of gymnastics said in the interviews:

“A coach that does not say much and who speaks a bit louder and slower for us older people, is much more effective, and that is the difference, since one should, as a coach, perhaps pay a little more attention to this contact with older customers.”

A participant of a soccer class emphasised the necessity of coaches adapting their behaviour to situational contexts: “... it always depends on the situation. Sometimes I feel under pressure and sometimes it helps me [if the coach gives me instructions]. The coach has to recognize that.”

This behaviour-related competence—organisational competence—is primarily related to the sport and exercise classes’ content and team sports’ content. The organisational competence comprises the coordinative aspects of sport and exercise classes and team sports in order to ensure a smooth and efficient process when exercising. For example, a basketball player said in the interview: “Because they are thinking about what we need, how to train, where and when we should be and the whole organization. That is why, I find it very important that you have a coach who cares about these things.” A soccer class participant defined organisational competences of a coach in this way: “…then the trainer says we have to do this and then that. He gives us specific instructions on what to do… and he explains how the training session is structured and how it works.” It is particularly evident in team sports that a relaxed training atmosphere is important as well as having structured training aspects, such as a warm up, a tactical part, a technical part and a fun part. This was deemed desirable to players. In teams where the organization of sport and exercise classes was unstructured, it resulted in confusion and dissatisfaction of the players and even (the worst-case scenario) injury.

### 3.2. Interactions of Coach Competences in the Integrated Model

Inducing positive AR in sport and exercise is a behavioural process which is characterised by the interaction of the aforementioned competences. At the beginning of this process, context sensitivity and empathy are part of the situations’ evaluative process. The evaluation process implies that the coach recognizes and analyses the situation. The coach recognizes when and where he/she has to deal with something, and with whom he/she has to deal with. The product of this analysis has a huge influence on the selection of the sport and exercise classes’ type (professional competences) and which behavioural strategies are required (social–emotional competences). The more extensive the level of the coach’s professional competences is, the bigger the repertoire of available content. A coach with high professional competences in his/her behaviour has the advantage of choosing from a bigger pool of potential exercises. It gives him the possibility to demonstrate the required skills better (motor competence), to give more precise instructions and correct respectively as well as control the process in sport and exercise classes more efficiently. A coach with high social–emotional competences benefits from a larger variety of behavioural and communicational strategies. In the interviews, it was often stated that with a bit of ‘feeling’ (empathy/relationship management) a coach can explain the exercise or movements better. This statement was made in all different sports, exercise groups and team sports.

Another important aspect of competences was to adapt the communication strategies as well as contents of training to the social context and demands of the training situation. On the one hand, this means that the coach must adapt the behavioural strategies used in that specific context and anticipate the effects his/her behaviour might have on the participants. On the other hand, the coach must adapt the training contents to the fitness and ability level of athletes (adaptability competence). 

It was found that certain categories act as regulators for other categories. In this sense, context sensitivity moderates the influences of all other components of the model and has a significant effect on the coach’s behaviour. For example, the over 60-year-old female participants in the gym class group stated that for them it was important that the coach “lent an ear” to the participants, if they had specific problems related to their sex and age, and adapted the exercise according to their specific needs (see also the citation in the paragraph empathy). The coach has the specific task of adapting sport and exercise class requirements to older women (adaptability competences) and should have the professional competences to make for an efficient and satisfying group training session for the participants. In these sport and exercise class sessions the coach must recognize the individual needs of the female participants. He/she must react to them. Therefore, the coach needs to know how his/her behaviour is perceived by the participants. Based on the statements in the interviews, it can be assumed that the effects of professional competences as well as social–emotional competences base on the elements of context sensitivity. The coach can use his/her professional competences and social–emotional competences optimally when he/she recognizes the requirements of the situation (context sensitivity). 

The relationship between professional competences and social emotional competences also showed up on the basis of the interview statements as important. This combination is primarily represented in the category as behaviour-related competences in the category adaptability competence (Figure 1). The specific expertise is selected and individually adapted to the respective participant. This participant input is then appropriately communicated and applied. The category ‘social–emotional competences’ interacts with the category ‘professional competences’ when the coach communicates the sport-specific know-how to the participants. Whether the athlete really receives the content of the instruction (professional competence) depends on the relationship between coach and athlete and on the coach’s social–emotional communication strategies (social–emotional competences). This became evident in answers to questions relating to exercises in which the coach demonstrated movement sequences and when the coach corrected the actions of athletes. Multiple mentions were made in the course of the interviews. It appeared that of the 18 interviewees, 16 participants mentioned empathy, just as many motivation and all participants mentioned aspects of context sensitivity.

Motivation competence is a behaviour-related competence which is primarily based on social–emotional competences. Primarily, relationship management is an important determinant of motivation competence, because a good relationship is a prerequisite for motivational processes. As part of the motivation competence, the coach has the task to motivate the participants individually, depending on the situation, and the specific characteristics and possibilities of the person. Thus, motivation competence interacts closely with context sensitivity, social–emotional competences and professional competences. The communicative aspects (verbal/nonverbal) of the coach’s behaviour with the participants play an essential role here. The participants often mentioned aspects of motivation in certain situations as one of the key competences of a coach. In these situations, the coach communicates with the participants in a positive and encouraging way. This was an important aid for the participants to be able to manage the exercises or movements. It brings the participants the satisfaction of moving, even during difficult exercises or sporting environments. These aspects of communicated motivation in sport and exercise classes meant that in some instances the interviewees trained and exercised regularly. 

Organisational competence bases primarily on professional competences and interacts closely with context sensitivity and the categories of social–emotional competences relationship management and empathy.

## 4. Discussion

The main objective of this work was to examine, from a participant’s perspective, which behavioural characteristics from the coaches are desired, in order to answer the resulting question: How do coaches induce positive AR in participants of sport and exercise? Drawing on a qualitative analysis of interviews with 18 participants, an integrated model (Figure 1) was developed to describe specific and general competences of coaches to induce positive AR. 

### 4.1. General Competences

The integrated model (Figure 1) included three different general competences as foundation for three specific competences of coach behaviour in sport and exercise.

The exceptional importance of context sensitivity became evident in the interviews. Davidson and Begley [18] used the term ‘context sensitivity’ to explain how well emotional reactions can be adapted to the social context. In their view, context sensitivity is one of the six dimensions of emotional styles. For Davidson and Begley [18] context sensitivity is a matter of intuition and cannot be intentionally managed. The social context as well as the own behaviour often follow an emotional subtext. In the present study, context sensitivity acts as a perception competence, which helps to assess the situation and the social context. According to the Coaching Model, coaches who possess high context sensitivity adapt their behaviour to the situation’s requirements accordingly [25].

Social–emotional competences are key competences for coaches and constitute a prerequisite for the behaviour-related competences. Three aspects of social–emotional competences were identified: empathy, self-management and relationship management. These general competences have also been identified as components of emotional intelligence (e.g., Boayatzis et al. [30] and Salovey & Mayer [31]). Martin and Fox [26] showed two types of leadership styles, which have been termed as bland or enriched [24,25,26]. In the present study, similarities to components of the model are noted. The enriched leadership style shows many similarities to several aspects of the social–emotional competences, mentioned in this study. This is similar to the category of the social–emotional competences in which the coach perceives the needs of participants. An enriched leadership style is characterised by social interaction and pleasantness, which might be a consequence of high social–emotional competences [26]. The first subcategory of social–emotional competences is empathy. It is closely connected to context sensitivity and is also a perception competence, which assesses the specific personal situation individually. Krug [32] found that ‘empathy’ and ‘sensitiveness for the specific context’ are desirable of coach competences. This aspect is confirmed in this study. This is clearly reflected in the number of mentions of ‘empathy’ in the interviews. The second subcategory is self-management. It enables the coach to organize himself and manage his/her own emotional states. A coach that is not enthusiastic himself cannot convey enthusiasm to participants in the training. Lee, Wäsche and Jekauc [33] found that the key for success for football coaches was the competence to regulate own emotions. The competence to regulate own emotions affects self-confidence, concentration and openness for social interactions of football coaches and the team. The coach needs to know in which way he/she wants to act, behave and develop as a leader, and which values he/she represents [34]. The third subcategory is relationship management. Relationship management aims to create a partnership between the participant and the coach. This partnership is based on the communication (verbal/nonverbal) between coach and participant, the leadership of the coach and the cooperation (verbal/nonverbal) between the coach and the participant [32]. 

Another prerequisite of general competences for coach behaviour are the professional competences. Motor competence and professional expertise were described by many participants, especially factors such as comprehensive education and extensive experience. This experience and knowledge ought to have been developed in their career as a coach, but also ought to have been acquired during their own athletic career. These two aspects were often named in the interviews, regardless of the type of sport. This approach was also suggested by Werthner and Trudel [35] in which the coach acquired knowledge through mediated (e.g., attending clinics), unmediated (e.g., observing other coaches) and internal (e.g., reflecting on their experience) learning situations. However, Carter and Bloom [36] found in their study that there was a lot of variation in the knowledge of a professional coach and that, for example, those with a lack of athletic experience purposely worked harder to either acquire or overcome any gaps in their knowledge. These professional competences are not that important for the training with beginners, because beginners cannot evaluate the quality of the content of sport and exercise classes and the competence of the coach [22]. However, the dropout rates are highest, especially at entry level in sport. In this context, the professional competences enables higher efficiency of the sport and exercise classes for the participant: As professional competences allow coaches to apply theory in their practice, competences become an important part of the coaching process and must be thoroughly understood in order to enhance coaching effectiveness [7].

### 4.2. Behaviour-Related Competences

The resulting model is based on the experiences and perspectives of the interviewed participants, who made statements on different kinds of behaviours, coach competences and sport and exercise class content, which in several ways interact with each other. The sum of this behavioural content, sport and exercise class content and context sensitivity results in the behaviour-related competences. These specific competences are the immediate determinant of coach’s behaviour and allow the coach to induce positive AR. 

In the category motivation competence of the behaviour-related competences, the communicative aspects (verbal/nonverbal) in the coach’s behaviour are very important. In accordance with Strauch et al. [22] interviewees very often named motivational factors as key coach competences. Communication with the coach brings satisfaction and security to participants, especially during difficult training situations, and can also motivate to accept challenges in their sport. Like in the interviews by Strauch et al. [22], the participants of this study confirmed that they come to training more regularly or more frequently due to a good motivation by the coach.

The category adaptation competence is the most complex category of the behaviour-related competences and interacts strongly with context sensitivity. Based on the specific context, special knowledge is selected, adapted to the participants and individually communicated. Most interviewed participants mentioned this ability as a central element for the coach’s behaviour. Professional competences allow the coach to apply theoretical knowledge in practice. However, additional behavioural competences are necessary to translate the knowledge into practice [37]. Coaches produce a specifically adapted training situation and thus induce positive AR within the participants. Therefore, coaches in team sports pay more attention to their leadership style and to their self-control than those in individual sports. This is mainly due to the high number of athletes and the heterogeneity in the group. In individual sports, the coach has more time for a closer and more personalised care. He/she also has more time for individual correction and explanatory statements.

The category organisational competence of the behaviour-related competences is not that important in individual sports as it is in team sports and exercise classes. Based on the larger number of participants and the different needs in team sports, the coach should be able to organize the different settings in sport and exercise.

Especially in the early stages of exercise maintenance, the interactions of coach competences, primarily the behaviour-related competences, could influence dropout rates, for example, during the first six months of sport and exercise classes. Our model shares parallels with the self-determination theory [38]. Klain et al. [39] showed in an empirical study that basic psychological needs were significantly higher in cases of closer and more personal contact between coach and participant than in the case of impersonal relationships between the coach and the participant. Furthermore, Wienke and Jekauc [6] suggest that psychological needs, according to the self-determination theory, are also influencing factors of AR in sport and exercise. There are aspects of social–emotional competences as well as the motivational part in the behaviour-related competences which might be related to the psychological needs in the self-determination theory. It might be that coaches with well-established social–emotional competences have the ability to address the psychological needs of participants, like competence, social relatedness and autonomy. The importance of motivation was also shown in a study by Olympiou, Jowett & Duda [23]. The results of this study support the notion that the coach–athlete relationship has implications for the motivation of participants in sport and exercise. According to Wienke and Jekauc [6], perceived physical exertion is another determinant of positive AR in sport and exercise. If participants are moderately exerted during sport and exercise, positive AR can be expected [40]. It can be supposed that coaches with high professional competences have the ability to properly dose the intensity of the sport and exercise classes to achieve a medium level of exertion in participants. Thus, a link between positive AR and professional competences might be the key to knowing how to dose the intensity of the sport and exercise classes to achieve moderate exertion afterwards. Context sensitivity and empathy might additionally contribute to knowing how and when to apply the professional competences. It can be assumed that the competences can implicitly be acquired through many years of experience without special education. By observing the reactions of participants, the coach has the possibility to obtain a feedback about his/her behaviour and sport and exercise class content. In this context, experiential knowledge and informal education are of great importance [7]. Chelladurai [24] assumed in the Multidimensional Model of Leadership that the coach’s behaviour influences the satisfaction and performance of the participants. The corresponding coach’s behaviour is a result of the specific conditions of the situation, characteristics of the coach and the participants characteristics [41]. Salmela and Russel [42] suggest in their Coaching Model that coaches construct a mental model of their athletes’ and teams’ potential. This mental model dictates how the coach applies the primary components of organization, training and competition to their athletes. Also, it is influenced by three components: the athlete’s characteristics, the coach’s characteristics and the factors of the specific context. These three components are also reflected in the present model of coach competences. These components were summarised in the behaviour-related competences, but they are also identifiable in the subcategories of the present model. All elements of the model are important for a coach to trigger positive AR within the participant.

### 4.3. Strengths and Restrictions

A strength of this study is the systematic analysis and interpretation of the interviews, which uses principles of the Grounded Theory. In this way, an in-depth understanding of how coach competences can induce positive AR in sport and exercise became feasible. Another strength of this study is the large heterogeneity of the interviewees in terms of sex, age and type of sport. However, qualitative studies of this kind are also restricted in various ways. Since the aim of this study was not to verify hypotheses, but to generate new insights or rather new hypotheses that can be drawn from the postulated model, the findings are not easily generalizable. The results are context-specific and will have to be tested in different contexts and with larger samples to enlarge their scope. Due to the selection of the interviewed participants, the study is delimited to the context of an urban area in Central Europe. Studies in other regional contexts with different sociocultural, legal and economic conditions might come to different conclusions. Moreover, the study is delimited by a focus only on adults. Children (aged 0–10) and adolescents (aged 11–17) were not analysed. Furthermore, the interviewed participants were involved in recreational sports; hence, the results cannot be transferred easily to the field of professional sports. The findings of qualitative data are always dependent on the knowledge and interpretation of the researchers involved. While the findings are based on an empirical analysis and allow an in-depth understanding, different interpretations might be possible.

### 4.4. Future Studies 

In future studies, the postulated model needs to be tested in different contexts, such as health sports programs, commercial leisure activities, fitness clubs, etc. Therefore, instruments for quantitative measurement need to be developed. Following this, the assumptions and the structure of the model developed in this paper could be empirically tested. In another step, the predictive validity of the model could be investigated in a longitudinal study. In this way, it could be analysed whether coaches with higher context sensitivity, social–emotional competences, professional competences as well as behaviour-related competences have a higher propensity to increase compliance for participation in sport and exercise courses than coaches with lower competences. 

### 4.5. Implications

The findings of this research can be helpful in various practical contexts, such as health sports or recreational sports. These results can contribute to different contents (e.g., motivational or instructive behaviour of coaches) and emphases in the education of coaches of various sports types, such as team sports or individual sports. Currently, sports education is based largely on the respective sport-specific expertise and less on the sport-specific individual social–emotional competences. Hence, coach education can be broadened by teaching social–emotional competences. The findings can also help develop criteria to assess coaches in different contexts [43], for example, quality management in a sports facility or the recruitment of new coaches. These arrangements would optimize the quality management in the specific workplace of coaches and of companies that employ coaches. Furthermore, the results of this study can help optimize the selection of coaching staff for companies and enable the possibility of a targeted assessment centre. In addition to examining formal qualifications of coaches, motivational, organisational and adaptive competences need to also be examined. Krug [32] also noticed that the staff development of coaches should focus on the personality and the individual key characteristics of each coach. 

## 5. Conclusions

This qualitative study provides a model of coach competences, which explain how positive AR in participants of sport and exercise can be induced. The model comprises six key competences for coaches that are responsible for coach behaviour in various sport contexts. The qualitative data in this study suggests that behaviour-related competences are crucial to induce positive AR. These behaviour-related competences are based on social–emotional competences, professional competences and context-sensitive abilities in dealing with the participants. An application of this model could be used to develop interventions and educational strategies to increase positive emotions, motivation as well as regular participation in sport and exercise courses.

## Figures and Tables

**Figure 1 sports-07-00016-f001:**
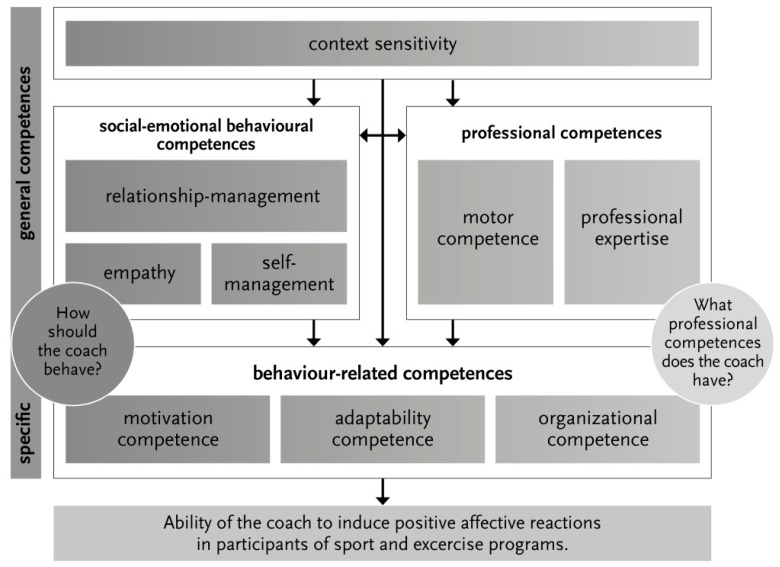
Integrated model of coach competences.

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
