# Peer review of "Coach Competences to Induce Positive Affective Reactions in Sport and Exercise—A Qualitative Study"

_sports, 2019, doi:10.3390/sports7010016_

Round 1
Reviewer 1 Report
Please find my comments attached in the pdf.

Author Response
Response to Reviewer 1 Comments
Thank you for your explanations in the pdf and the helpful comments.
Point 1: are
Response 1: Thank you for this comment. We checked the sentence and think the word “are” is correct in this case.
Point 2: Please add Standard Deviation.
Response 2: Thank you. The standard deviation has been added.
Point 3: This reference is very old. Please add these two references to this statement to strengthen it (https://www.mdpi.com/1010-660X/54/5/84;
https://link.springer.com/article/10.1186/s40064-016-2033-8)
Response 3: Thank you for this aspect and the important references. We added the two recommended references to the manuscript to support our
statements. Furthermore, we removed old references (amongst other things Dishman, R.K.; Buckworth, J. Increasing physical activity: A quantitative
synthesis. Medicine and science in sports and exercise 1996, 28, 706-719.) and added new ones (amongst others Wienke, B.; Jekauc, D. A qualitative
analysis of emotional facilitators in exercise. Frontiers in Psychology 2016, 7.).
Point 4: In the same city?
Response 4: Thanks for this question. Yes, the study was done in the same city (Berlin, Germany’s capital with more than 3.5 million inhabitants). We
have added this aspect to the manuscript to clarify the context of the study.
Point 5: The average age in the abstract is different.
Response 5: Thank you for this note. We corrected the average age (42,6) in the abstract.
Point 6: …the university of X (anonymized for review).
Response 6: Thanks for the advice. We thought it was an anonymous review and therefore did not mention the university and the ethics committee. We
have included the information and the study has been accepted by the ethics committee of the Goethe University of Frankfurt/Main (code 2018-51).
Point 7: Please describe this training.
Response 7: Thank you, we elaborated on this aspect. During the training with the interviewers, techniques for semi-structured interviews, in which
participants have room left for their own thoughts and comments, were trained.
Point 8: I recommend you to discuss your results also with this paper:
https://revistas.um.es/sportk/article/view/293611
Response 8: Thank you for the hint. We added this interesting reference concerning the further development of coaches to the manuscript.
Point 9: …tested in different contexts - for example?
Response 9: This is a good hint. With different contexts we mean different sport environments. For example, health sports programs, commercial leisure
activities or fitness clubs.
Point 10: and
Response 10: Thanks, we changed it accordingly.
Point 11: …helpful in various practical contexts - for example?
Response 11: Thanks for the comment. The various practical contexts can be programs in either health sports or recreational sports.
Point 12: ... contribute to different contents - for example?
Response 12: Thank you. The results of this study can contribute to different contents, such as to motivational or instructive coach behaviour.
Point 13: … coaches of various sports types - for example?
Response 13: Thank you. The results of this study can be used in educating coaches of various sports types, such as in team sports or individual sports.
Point 14: …coaches in different contexts - for example?
Response 14: Thanks for the advice. The findings of this study can help develop criteria to assess coaches. For example, quality management in a sports
facility or recruiting new coaches.
Point 15: I suggest to add dois.
Response 15: Thank you. We have added the recommendation and other references to the manuscript.
Point 16: … for canadian adults.
Response 16: Thanks, we have corrected it with the capital letter “C” to “Canadian adults”.
Reviewer 2 Report
Dear author, I found your manuscript really interesting.
Here are some suggestions for improvement:
Introduction
It is appropriate, however, many of the quotes they use are very old, I recommend that you modify them for new ones, because what is raised in this
section is the current problem.
In addition, they must be careful with the forms of writing, because being a scientific article must be written in impersonal and not in personal (line 108).
Method
In the participants' section it says that the ethics committee was approved, although for the review this is anonymous, I think it should include the ethics
committee's code. Also, if the study were to be published, it would be necessary for the university to be included.
Results
I believe that this paragraph should be improved, as the first part is excessively short and it is not clear to me how these results have been achieved.
It also subsequently describes the competences of the Coach, this being a definition not part of the results. I believe that in order for these to form part of
the results, they must restructure this part.
Bibliographical references
We suggest that you check the bibliographical references, because in some of them I have seen an error, for example, in the abbreviations of the journals,
between the name of the journal and the year of publication there is a dot, between the year of publication and the volume there is a dot. The name of the
journal and the volume are cursive.
Author Response
Response to Reviewer 2 Comments
Thank you for your explanations, the nice words and the helpful comments.
Point 1: Introduction - It is appropriate, however, many of the quotes they use are very old, I recommend that you modify them for new ones, because
what is raised in this section is the current problem.
Response 1: Thank you for this important suggestion. We have removed older quotes (e.g., Mujika, I.; Padilla, S. Detraining: Loss of training-induced
physiological and performance adaptations. Part i. Sports Medicine 2000, 30, 79-87 or Dishman, R.K.; Buckworth, J. Increasing physical activity: A
quantitative synthesis. Medicine and science in sports and exercise 1996, 28, 706-719.) and have added new references (e.g., López-Sánchez, G.;
Emeljanovas, A.; Miežienė, B.; Díaz-Suárez, A.; Sánchez-Castillo, S.; Yang, L.; Roberts, J.; Smith, L. Levels of physical activity in lithuanian adolescents.
Medicina 2018, 54, 84; or Rabast, U. Gesunde Ernährung, gesunder Lebensstil: Was schadet uns, was tut uns gut? Springer-Verlag: 2018.).
Point 2: In addition, they must be careful with the forms of writing, because being a scientific article must be written in impersonal and not in personal (line
108).
Response 2: Thank you for the hint. We have replaced the personal forms with passive forms.
Point 3: Methods - In the participants' section it says that the ethics committee was approved, although for the review this is anonymous, I think it should
include the ethics committee's code. Also, if the were to be published, it would be necessary for the university to be included.
Response 3: Thank you for the suggestion. We thought it was an anonymous review and therefore did not mention the university nor the ethics committee.
Now we have included the information. The study has been accepted by the ethics committee of the Goethe University of Frankfurt/Main (code 2018-51).
Point 4: Results - I believe that this paragraph should be improved, as the first part is excessively short and it is not clear to me how these results have been achieved.
Response 4: Thank you for the advice. In the methods section, especially in part 2.3 on data analysis, we now describe in more in detail the working steps
of the qualitative analysis based on the Grounded Theory. In the third step of the Grounded Theory, the selective coding clarifies how the results of the
study emerged from the transcribed interviews:” In the final step, we applied selective coding, with a focus on the core category with its dimensions,
interactions (sub-categories) and definitions. This step includes a precise definition and explanation of the categories. Other categories that needed further
explanations were filled in with descriptive details. The selective coding was done by three researchers. Based on the identified competences and their
interactions, a model was constructed.” To further clarify our results, we added more quotations from the interviews.
Point 5: Results - It also subsequently describes the competences of the Coach, this being a definition not part of the results. I believe that in order for these to form part of the results, they must restructure this part.
Response 5: Thanks for the comment. As described in the methods section, the categories and definitions of the categories have been developed from the
interview content using the Grounded Theory approach. The categories were defined and explained based on the participants’ statements during the
interviews. Therefore, the definitions are part of the study’s results. However, we acknowledge that that it is not clear enough and that the definitions of the
categories had been developed from the analysis of the interviews. We addressed this issue in the manuscript. The structure of presenting the categories
always includes a definition of the category, followed by an explanation and one or more examples from the interviews. This approach is common in
qualitative research and we considered it to be suitable and useful also in our case.
Point 6: Bibliographical references - We suggest that you check the bibliographical references, because in some of them I have seen an error, for
example, in the abbreviations of the journals, between the name of the journal and the year of publication there is a dot, between the year of publication
and the volume there is a dot. The name of the journal and the volume are cursive.
Response 6: Thank you for the hint. We checked all references and corrected them. We have now used the abbreviations of the journal names in almost
all sources, and have supplemented the dot between the name of the journal and the year of publication. As dictated by the citation guidelines of ACS
style, we have added a comma between the year of publication and the volume. We checked all journals, if the name and the volume were cursive or not.
In the books only the titles are cursive and in the thesis nothing is written in cursive.
Reviewer 3 Report
The article was carefully written, and the research was carried out reliably. I only see minor things to be corrected (line 538 und / and). The article in my
opinion is suitable for publication because it raises an important issue. Some insufficiency leaves only a small number of respondents' answers, based on
which the conclusions were drawn. Publication in electronic form can be long also if the authors would like to post even more statements of respondents,
that would be interesting for the reader. Perhaps the authors would also be able to describe the socio-cultural context in which the research was carried
out. Undoubtedly, such analyzes will be important in the future not only among children but above all in multicultural society. Leading a sports group
composed of people from different cultures, whether at school or in an international corporation, will be a challenge for people running sports activities.
Different people may have different expectations towards their sports leader.
Author Response
Response to Reviewer 3 Comments
Thank you for your explanations, the nice words and the helpful comments.
Point 1: I only see minor things to be corrected (line 538 und / and).
Response 1: Thank you. We corrected the error.
Point 2: Publication in electronic form can be long also if the authors would like to post even more statements of respondents that would be interesting
for the reader.
Response 2: Thanks for this comment. We agree and added more statements from the participants to clarify the categories.
Point 3: Perhaps the authors would also be able to describe the socio-cultural context in which the research was carried out.
Response 3: Thank you for the suggestion. Besides age and gender, we did not collect any further parameters such as social class or religion of the
participants. However, we added some more information concerning the socio-cultural setting of the study, which took place in Berlin, the capital of
Germany with more than 3,5 million inhabitants. We included this aspect as a limitation in the section “Strengths and Limitations”, because then other
socio-cultural parameters can be included in subsequent studies.
Round 2
Reviewer 2 Report
The changes have been made correctly. Even so, some quotations from the journal "sport" itself could be included, as there are articles that could be of
great scientific interest to the authors.
Author Response
Response to Reviewer 2 Comments
Thank you for your explanations and the helpful comments.
Point 1: The changes have been made correctly. Even so, some quotations from the journal "sport" itself could be included, as there are articles that
could be of great scientific interest to the authors.
Response 1: Thank you for this suggestion. We've added quotes from the journal “sport” itself (e.g., Kopp, A.; Jekauc, D. The influence of emotional
intelligence on performance in competitive sports: A meta-analytical investigation. Sports 2018, 6, 175.; Lee, H.; Wäsche, H.; Jekauc, D. Analyzing the
components of emotional competence of football coaches: A qualitative study from the coaches’ perspective. Sports 2018, 6, 123.)